# Neutrophil to Lymphocyte Ratio as Prognostic and Predictive Factor in Breast Cancer Patients: A Systematic Review

**DOI:** 10.3390/cancers12040958

**Published:** 2020-04-13

**Authors:** Iléana Corbeau, William Jacot, Séverine Guiu

**Affiliations:** 1Department of Medical Oncology, Institut du Cancer de Montpellier (ICM), 208 Avenue des Apothicaires, 34 298 Montpellier, France; william.jacot@icm.unicancer.fr (W.J.); severine.guiu@icm.unicancer.fr (S.G.); 2IRCM, INSERM U1194, Univerité de Montpellier, ICM, 208 Avenue des Apothicaires, 34 298 Montpellier, France

**Keywords:** breast cancer, inflammatory blood markers, neutrophil to lymphocyte ratio, prognostic factor, predictive factor, pathological complete response, toxicity

## Abstract

Inflammatory blood markers (IBM), such as the neutrophil to lymphocyte ratio (NLR), have emerged as potential prognostic factors in various cancers, including breast cancer (BC), potentially allowing an easy, minimally invasive evaluation of a given cancer‘s prognosis and treatment outcome. We report here a systematic overview of the published data evaluating NLR as a prognostic factor or predictive factor for pathological complete response (PCR) and toxicity in early and advanced BC. A total of 45 articles were identified. NLR was found to be an independent prognostic factor for survival in most of the adjuvant treatment studies. However, no significant correlation was found between survival and NLR for early BC patients receiving neo-adjuvant chemotherapy (NACT) and advanced BC patients. Most studies failed to find a significant correlation between NLR and PCR after NACT. Finally, some data showed that IBM could be predictive of chemotherapy-related toxicity.

## 1. Introduction

Breast cancer (BC) prognosis depends not only on the tumor stage (localized versus metastatic disease), but also on the molecular subtype (luminal, HER2+, or triple-negative BC). Currently, BC management includes multidisciplinary and multimodal treatments: surgery, radiation therapy, chemotherapy, endocrine therapy and/or targeted therapies [1,2]. Although some predictive and/or prognostic factors are available (for example, hormone receptor status, HER2 overexpression/amplification, histological grade or stage), additional predictive and prognostic biomarkers are needed to better adapt the treatment to each individual patient.

In the last few years, inflammatory blood markers have emerged as predictive and prognostic factors, particularly the neutrophil to lymphocyte ratio (NLR); that is, the ratio between the absolute neutrophil count and the absolute lymphocyte count. Lymphopenia and a high NLR before chemotherapy initiation have been inconsistently associated with worse responses to neo-adjuvant chemotherapy and with poor prognoses of different cancer types, including BC [3,4]. 

The role of inflammation in cancer is now well established [5], and has been described at different stages of cancer development (initiation, promotion, invasion, and metastasis). Activated inflammatory cells are sources of reactive oxygen species and reactive nitrogen intermediates that can induce DNA damage and genome instability, thus promoting cancer initiation [6,7], or interfere with the DNA repair systems [8]. Inflammation increases the production of growth factors and cytokines that can confer a stem cell-like phenotype to tumor progenitors. Inflammation also promotes upregulation of angiogenic factors (known as angiogenetic switch) that favor tumor progression. 

Neutrophils have been the focus of much research, and there is now evidence that they can promote tumor growth and play a role in metastasis development [9,10]. Their ability to secrete proteases, particularly matrix metalloproteases, contributes to favor a tumor’s invasion. Neutrophils are also involved in tumor progression through their capacity to activate signal transducers and activators of transcription 3 (STAT3) and to promote neo-angiogenesis [11]. 

In recent years, the role of tumor infiltrating lymphocytes (TILs), especially in BC, has also been studied [12]. TILs are a selected population of T cells that show a high specific immunological reactivity against tumor cells. These lymphocytes, which are part of the innate immune system, can detect cancer cells and alert the immune system that will destroy them. Therefore, a low TIL count could be predictive of a lower response to neo-adjuvant chemotherapy [13], and might be associated with poor prognosis [14,15].

Peripheral inflammatory blood markers could be helpful for predicting patients’ prognoses and also their response to neo-adjuvant chemotherapy in BC. Many studies evaluated the NLR, with conflicting results about its value as a predictive and/or prognostic factor [11,16,17,18,19,20,21]. Studies on the total white blood cell count, neutrophil count, lymphocyte count and other blood cell ratios, such as the platelet to lymphocyte ratio (PLR), also gave inconsistent results. 

Here, we systematically collected published data on the predictive or prognostic role of NLR in patients with BC. 

We first summarized the published data on NLR and treatment efficacy, evaluated in terms of disease-free survival (DFS), breast cancer-specific survival (BCSS) and overall survival (OS), in patients with early BC (treated with neo-adjuvant or adjuvant chemotherapy) and with advanced BC. We also summarized the data on the predictive value of NLR for pathological complete response (PCR). 

Then, we evaluated the correlation between chemotherapy-related toxicity and NLR and/or lymphopenia. 

## 2. Material and Methods

### 2.1. Search Strategy

We performed a systematic search of the PubMed database using the following search terms: “neutrophil to lymphocytes ratio” or “lymphopenia” AND “breast cancer”. We also looked for articles using the search terms “toxicity” AND “neutrophil to lymphocyte ratio” AND “breast cancer” and “toxicity” AND “lymphopenia” AND “breast cancer”. We updated the literature search in February 2019. We also screened the references of the selected articles found in PubMed in order to ensure exhaustivity. We followed the PRISMA guidelines throughout the process. 

### 2.2. Study Selection

For the first part on NLR and treatment efficacy, we included all articles with only BC cohorts and PCR and/or survival analysis as the primary objectives. We excluded articles that concomitantly assessed different conditions (e.g., neo-adjuvant, adjuvant and metastatic settings). For the second part on toxicity, we included all articles reporting data on BC, inflammatory blood markers and chemotherapy-related toxicity. For both parts, we excluded all articles that were not in English (Figure 1). 

### 2.3. Data Extraction

Two reviewers independently extracted the following data from the selected studies: name of the first author and year of publication, population of interest, description of endpoints based on BC molecular subtypes, number of enrolled patients, ethnicity, treatment received, chosen NLR cut-off, primary objective and results (univariate analysis), results of subgroup analyses (if applicable), secondary objectives and their results, and multivariate analysis of NLR results (multivariate models and covariates used for adjustment). The two reviewers distinguished three populations: patients receiving neo-adjuvant chemotherapy, patients receiving adjuvant chemotherapy and patients with advanced BC. 

For the studies on toxicity, the two reviewers selected articles on chemotherapy-related toxicity in the function of the lymphocyte count or NLR, and collected data on the population of interest, received treatments and toxicity. 

If the selected articles also reported data on other inflammatory blood markers (e.g., lymphocyte count or PLR) as predictive and/or prognostic factors, we included this information in our tables.

### 2.4. Definitions

DFS was defined as the time from diagnosis (or the date of surgery for patients receiving adjuvant chemotherapy) to the date of relapse (local recurrence or metastases to distant sites) and/or death from any cause. In some papers, recurrence-free survival (RFS) was the primary objective. As RFS has the same definition as DFS, we used DFS for both DFS and RFS in this work. 

BCSS was calculated from the date of diagnosis to the date of death by cancer or of the last follow-up visit. Some articles reported data on disease-specific survival (DSS), which has the same definition as BCSS. Therefore, we used the term BCSS for both BCSS and DSS in this work. 

OS was defined as the time from the date of diagnosis (or the date of surgery) to the date of death due to any reason, or the date of the last follow-up (for some papers). 

Progression-free survival (PFS) was defined as the time from treatment initiation to the date of disease progression, or death from any cause.

## 3. Results

### 3.1. NLR and Treatment Efficacy

#### 3.1.1. In Early BC

##### (1) Patients Receiving Neo-Adjuvant Therapy

Ten studies reported data on NLR as a predictive or prognostic factor in patients receiving neo-adjuvant therapy [22,23,24,25,26,27,28,29,30,31]. Among these studies, eight included patients with all molecular subtypes [22,23,24,25,26,27,28,29], one included only patients with triple negative BC (TNBC) [31] and the last one included only patients with hormone receptor-positive/HER2-negative BC [30]. The NLR cut-off for the statistical analyses was between 1.7 and 3.33, and was computed by a receiver operating characteristic (ROC) curve analysis in all studies. The number of enrolled patients ranged from 78 to 373. Five studies reported data on Asian populations [23,25,28,30,31]. In nine articles, the neo-adjuvant therapy protocols were described, and were based on anthracycline and/or taxane [23,24,25,26,27,28,29,30,31]. (Table 1).

###### Results on NLR and PCR

Studies including All BC Molecular Subtypes: the PCR rate in the function of NLR was the primary objective in seven studies [22,23,24,25,27,28,29]. In all studies (except for two without definition), PCR was defined as the complete disappearance of invasive tumors in breast and lymph nodes (patients with residual ductal carcinoma in situ were also considered to have achieved PCR: ypT0/is pN0). Among these seven studies, only three (43%) found a significant correlation between NLR and PCR in the univariate analysis [23,25,28]. Only Qian et al. analyzed data using a multivariate model and found that NLR was not an independent prognostic factor for PCR (*p* = 0.254) [28].

Studies including Only Specific BC Molecular Subtypes: Chae et al. did not find any relation between NLR and PCR in the univariate and multivariate models (odds ratio (OR) = 4.274; 95%CI 1.451−12.658; *p* = 0.008) [31] in 87 patients with TNBC. 

Subgroup Analyses: Suppan et al. [24] did not find any correlation between NLR and PCR in patients who received both anthracyclines and taxanes and in patients who received only anthracyclines or taxanes. They also compared the different BC molecular subtypes, and did not identify any correlation between NLR and PCR in any subtype. 

Graziano et al. [27] showed that PCR rates were higher in patients with NLR^low^/PLR^low^ than patients with NLR^high^/PLR^high^ (OR = 1.98; 95%CI 1.01–3.89; *p* = 0.044).

###### Results on NLR and DFS

Studies including All BC Molecular Subtypes: DFS was the primary objective in six studies [23,24,25,26,28,29]. Only two (33%) found that a higher NLR was correlated with a shorter DFS in the univariate analysis [25,26]. Three studies (50%) analyzed data on NLR and DFS by multivariate analyses [24,25,26], and only one (33%) [25] showed that NLR was an independent prognostic factor for DFS (hazard ratio (HR) = 1.57; 95%CI 1.05–3.57; *p* < 0.05).

Studies including Only Specific BC Molecular Subtypes: Koh et al. [30] reported that in patients with hormone receptor-positive/HER2- negative BC, NLR was an independent prognostic factor for DFS (HR = 3.87; 95%CI 1.64–9.14; *p* = 0.002).

Sub Group Analyses: Asano et al. [23] showed that in the subgroup of patients with TNBC (*n* = 61), NLR was correlated with DFS in the univariate analysis, but was not an independent prognostic factor of DFS.

###### Results on NLR and OS and BCSS

Studies including All BC Molecular Subtypes: four studies evaluated NLR and OS by a univariate analysis [23,26,28,29], and only one (25%) [26] found a significant association between NLR and OS, but NLR was not an independent prognostic factor for OS (*p* = 0.543). On the other hand, Chen et al. [25] showed that in patients with stage II and III BC (*n* = 215), NLR was an independent prognostic factor for BCSS (HR = 2.21; 95%CI 1.01–4.39; *p* < 0.05).

Studies including Only Specific BC Molecular Subtypes: Koh et al. [30] showed that in 157 patients with hormone receptor-positive/HER2-negative BC, NLR was an independent prognostic factor for OS (HR = 24.87; 95% CI 3.1–201.3; *p* = 0.003).

Sub Group Analyses: Koh et al. [30] also found that the correlation between NLR and OS was stronger in patients with stage III BC than in patients with stage II BC.

###### Conclusion on NLR as Prognostic and Predictive Factor in Patients with Early BC Receiving Neo-Adjuvant Chemotherapy

Six articles reported the results of the multivariate analyses of the data on NLR and PCR, DFS or OS in patients receiving neo-adjuvant chemotherapy [24,25,26,28,30,31]. NLR was correlated with PCR in one article [31] (50% of *n* = 2), with DFS in two studies (50% of *n* = 4) [25,30], with OS in two articles (50% of *n* = 4) [30,31] and with BCSS in one article (100%) [25]. These analyses included a total of 1558 patients. The conclusion should be taken with caution because two of these six selected studies included only one molecular subtype (TNBC or hormone receptor-positive/HER2 negative BC). Table 2 summarizes the results of these multivariate analyses and the adjustment factors. 

###### Results on other Inflammatory Blood Markers

Among these studies, four [26,27,28,29] also evaluated the total lymphocyte count (*n* = 2) [26,28] or total neutrophil count before treatment (*n* = 2) [26,28], and other ratios, such as the lymphocyte to monocyte ratio (LMR) (*n* = 2) [26,29], neutrophil to monocyte ratio (NMR) (*n* = 2) [26,29] and PLR (*n* = 2) [27,29], as predictive and prognostic factors of survival. Lymphocyte count was not correlated with DFS and OS in the study by Marin Hernandez et al. [26], whereas it was an independent predictive factor for PCR in the multivariate analysis in the work by Qian et al. (OR = 4.37; 95%CI 1.43–13.39; *p* = 0.01) [28]. Neutrophil count was not correlated with PCR and OS in the studies by Qian et al. [28] and by Marin Hernandez et al. [26], respectively. Conversely, it was an independent prognostic factor for DFS (multivariate analysis) in the study by Qian et al. [28]. PLR was not correlated with PCR [27], DFS or OS [29] (univariate analyses). Similarly, NMR and LMR were not correlated with survival (DFS and OS) in the study by Marin Hernandez et al. (univariate and multivariate analyses) [26] and also in the study by Losada et al. (univariate analysis) [29]. 

##### (2) Patients Receiving Adjuvant Treatment

Twenty-three studies reported data on NLR as a prognostic factor in patients with localized BC receiving adjuvant treatment [32,33,34,35,36,37,38,39,40,41,42,43,44,45,46,47,48,49,50,51,52,53]. Most studies (*n* = 18; 78%) included patients with all BC molecular subtypes [32,33,34,35,36,37,38,39,40,41,42,43,44,45,46,47,48]. four included only patients withTNBC [49,50,51,52], and one work included patients with onlyhormone receptor-negative BC (independently of the HER2 status) [53]. The NLR cut-off for the statistical analyses, computed by the ROC curves analysis in 78% of these studies, was between 1.34 and 4. The number of enrolled patients varied from 90 to 1570. Fifteen studies (65%) reported data on Asian populations [32,35,36,38,39,42,43,44,46,47,48,49,51,53]. The adjuvant chemotherapy regimens were described only in six studies (anthracycline- and/or taxane-based regimens) [33,38,43,45,48,51]. (Table 3).

###### Results on NLR and DFS

Studies including All BC Molecular Subtypes: seventeen studies analyzed the correlation between NLR and DFS after adjuvant chemotherapy [33,34,35,36,37,38,39,40,41,42,43,44,45,46,47]. In the univariate analysis, NLR was correlated with DFS in 14/17 articles (82%) [34,35,37,38,39,40,41,42,44,45,46,46,47]. Among the 13 studies where multivariate models were used [34,35,37,38,39,40,41,42,44,46,47,48], only eight found that NLR was an independent prognostic factor for DFS [34,37,38,39,40,46,48]. 

Studies in Specific BC Molecular Subgroups: four studies reported data on patients with TNBC [49,50,51,52]. Three (75%) showed that a high NLR was correlated with a shorter DFS in the univariate analyses [49,50,51]. In the multivariate analyses, two studies found a statistically significant correlation between NLR and DFS in patients with TNBC receiving adjuvant chemotherapy [49,50]. 

In patients with hormone receptor-negative BC, Liu et al. [53] found a correlation between NLR and DFS in the univariate and multivariate analyses (HR = 1.89; 95%CI 1.42–2.51; *p* < 0.001).

Subgroup Analyses: many studies reported subgroup analyses based on the BC stage, molecular or pathological subtypes. Five studies tried to determine whether the results varied in the function of the molecular subtypes [32,38,39,44,46]. In two studies, NLR was correlated with DFS (and to DSS) only in patients with luminal cancer [44] and luminal A cancer [32]. Conversely, in two other studies, this correlation was observed only in patients with TNBC [38,39]. Another article reported that NRL was correlated with DFS in all molecular subtypes [46]. 

Moreover, in patients with lymph node invasion, NLR was correlated with DFS, but only in the univariate analysis [44].

Finally, two studies [45,46] showed that NLR was correlated with DFS in early stage BC (I and/or II according to the AJCC staging system), but not in stage III BC. 

###### Results on NLR and OS and BCSS

Studies including all BC Molecular Subtypes: among the 10 studies with data on the NLR and OS in patients receiving adjuvant treatment [33,34,36,37,38,39,41,42,45,47], six (60%) found a correlation in the univariate analysis [34,36,37,39,41,45]. Five studies analyzed the correlation between NLR and OS by a multivariate analysis [34,36,37,39,41], and four (80%) found that NLR was an independent prognostic factor for OS [34,36,37,39]. Three studies with data on the NLR and BCSS [32,35,44] showed that they were correlated (univariate analyses), and two (66%) identified NLR as an independent prognostic factor for BCSS [32,35]. 

Studies including Specific BC Molecular Subtypes: four studies assessed OS in the function of the NLR in patients with TNBC [49,50,51,52]. Three (75%) showed that a high NLR was correlated with a shorter OS (univariate analysis) [49,50,51]. They also found a significant correlation between NLR and OS (multivariate analysis) in patients with TNBC receiving adjuvant chemotherapy [49,50,51]. Liu et al. [53] found that in patients with hormone receptor-negative BC, a high NLR was associated with a poor OS in the univariate (HR = 3.09; 95%CI 2.35–4.06; *p* < 0.001) and multivariate analyses (HR = 3.09; 95%CI 2.35–4.06; *p* < 0.001). 

Subgroup Analyses: NLR was correlated with OS in the luminal A and TNBC subgroups in the study by Yao et al. [36], but only in the TNBC subgroup in the work by Jia et al. [39]. 

###### Conclusion on NLR as Prognostic Factor in Patients with Localized BC Receiving Adjuvant Chemotherapy

Twenty-three articles reported data on NLR as a prognostic factor in patients receiving adjuvant treatment [32,33,34,35,36,37,38,39,40,41,42,43,44,45,46,47,48,49,50,51,52,53]. Among the 29 multivariate analyses performed in these studies, 21 (72.4%) highlighted a positive correlation between NLR and survival [32,34,35,36,37,38,39,40,46,48,49,50,51,53], specifically DFS (11/17 analyses; 65%) [34,37,38,39,40,46,48,49,50,53], OS (8/9 analyses; 89%) [34,36,37,39,49,50,51,53] and BCSS (2/3; 66%) [32,35]. In total, 18,153 patients were enrolled in these studies. Table 4 summarizes the results of the multivariate analyses and the adjustment factors used.

###### Results on Other Inflammatory Blood Markers

Eleven studies reported results on other inflammatory blood markers. PLR was correlated with DFS in four [41,43,44,53] of the six (66.6%) studies that assessed this correlation [33,36,41,43,44,53], and with OS in two [41,53] of the four (50%) studies that focused on this question [33,36,41,53]. PLR was correlated with DSS in the study by Cho et al. [44]. Four articles [41,43,44,53] evaluated PLR in multivariate analyses. Three (75% of four) found that PLR was an independent prognostic factor for DFS [41,43,44], one (50% of two) found a correlation between PLR and OS [41] and one with DSS (100%; only one study) [44]. 

Two studies found that a derived NLR (dNLR, calculated as the ratio of neutrophils over white blood cells minus neutrophils) was correlated with DFS, OS and DSS in the univariate analyses [37,44], but not in the multivariate analyses [44].

LMR was correlated with DFS, OS and DSS (univariate analysis) in two [39,44] of the three studies that assessed this biomarker [39,43,44]. In the multivariate analyses, no correlation was found between LMR and survival. 

Among the four studies on lymphocyte count and survival [33,44,47,52], only one (25%) found that lymphocyte count was correlated with DFS and DSS [44], but never with OS. No multivariate analysis was performed. 

Finally, two studies [33,44] showed that neutrophil count, platelet count, and monocyte count were not correlated with survival. Cihan et al. [33] also reported no correlation between white blood cell count, eosinophil cell count, basophil cell count and survival (univariate analyses). 

#### 3.1.2. Patients with Advanced Breast Cancer

We identified eight studies with data on NLR as a prognostic factor in patients with metastatic BC [54,55,56,57,58,59,60,61]. Four articles (50%) enrolled patients with all molecular subtypes [54,56,57,58], two focused on patients with HER2-positive BC [55,61], one on patients with TNBC [60] and one on patients with hormone receptor-positive BC receiving hormone therapy as initial treatment [59](Table 5).

In these studies, the NLR cut-off value ranged from 1.9 to 3 (always based on previous studies), and the number of enrolled patients varied between 34 and 171. Six studies (75%) reported data on Asian populations [54,55,56,58,59,61].

##### (1) Results on NLR and PFS

Studies including All BC Molecular Subtypes: two studies reported data on NLR and PFS [56,57] and one (50%) showed that NLR was an independent prognostic factor for PFS (HR = 0.39; 95%CI 0.18−0.78; *p* < 0.001) [56]. 

Studies including only Specific BC Molecular Subtypes: Vernieri et al. [60] showed that in 57 patients with TNBC, NLR was an independent prognostic factor for PFS (HR = 2.65; 95%CI 1.36–5.18; *p* = 0.004). Two studies focused on patients with HER2-positive tumors [55,61], and one found that NLR was correlated with PFS in the multivariate analysis (HR = 0.27; 95%CI 0.10–0.63; *p* < 0.001) [61]. NLR was an independent prognostic factor for PFS in patients with hormone receptor-positive BC receiving endocrine therapy as initial treatment (HR = 3.93; 95%CI 1.4–10.84; *p* = 0.008) [59].

##### (2) Results on NLR and OS 

Studies including All Molecular Subtypes: among the three studies with data on NLR and OS [54,56,58], two (66%) [54] found that NLR was an independent prognostic factor for OS in patients with metastatic BC. 

Studies including Only Specific BC Molecular Subtypes: NRL was correlated with OS in patients with HER2-positive BC, in the univariate and multivariate analyses (HR = 0.35; 95%CI 0.13–0.84; *p* = 0.018) [61], and also in patients with hormone receptor-positive in the univariate analysis (multivariate results not available) [59].

##### (3) Conclusion on NLR as Prognostic Factor in Patients with Advanced BC

Six articles reported the results of the multivariate analyses on NLR and survival in patients with advanced BC [54,56,58,59,60,61]. All found a significant correlation between NLR and survival. However, such results should be considered with caution because the multivariate analyses did not concern all types of survival outcomes and the studied populations were very heterogeneous. Table 6 summarizes the results of the multivariate analyses and the adjustment factors used.

##### (4) Results for Other Inflammatory Blood Markers

Three articles had data on other inflammatory blood markers as prognostic factors for survival in advanced BC [55,60,61]. In the univariate analysis, PFS was correlated with the absolute lymphocyte count, PLR and LMR [55,60]; however, one study found no correlation between PLR and PFS [61]. Vernieri et al. [60] found that in the multivariate analysis, PLR was an independent prognostic factor for PFS. One analysis showed that PLR was not a prognostic factor for OS [61].

### 3.2. Toxicity

We identified four articles that reported data on inflammatory blood markers and treatment-related toxicity [62,63,64,65] as the primary objective. Only one study analyzed the correlation between NLR and toxicity in a cohort of patients with BC (Yamanouchi et al. [65]). The three other studies focused on lymphopenia (defined as lymphocytes value < 0.7 G/L) in a cohort of patients with different cancer types (including BC). Two studies evaluated a French population [62,63], and the other two concerned Asian patients [64,65]. (Table 7).

Ray Coquard et al. [62] evaluated the predictive factors for early death after chemotherapy (defined as death within one month after the administration of cancer treatment) in a prospective study. They included 1051 patients among whom 756 (33%) had BC. They found that lymphopenia was a predictive factor of worse survival, in the univariate and multivariate models (OR = 3.1; 95%CI 1.8–5.8; *p* < 0.001). In a successive article, the same authors [63] determined whether a lymphocyte count <0.7 G/L at day 1 was predictive of febrile neutropenia in three cohorts of 950, 321 and 329 patients (including the cohort used in the previous work) among whom 24%, 33% and 42% had BC, respectively. They found that in the largest cohort, a lymphocyte count <0.7 G/L at day 1 was predictive of febrile neutropenia in the univariate and multivariate analyses (OR = 1.75; 95%CI 1.49–4.8; *p* = 0.02).

A Korean study [64] on 82 patients (including 11 (13%) patients with BC) did not find any correlation between lymphopenia at day 1 of the first chemotherapy course and risk of febrile neutropenia. However, due to the small number of patients with BC, no definitive conclusion could be drawn for this population.

Finally, Yamanouchi et al. [65] did not find any correlation between peripheral neuropathy occurrence and NLR, PLR or MLR in a cohort of 67 patients with BC who received at least four cycles of docetaxel (75 mg/m^2^).

## 4. Conclusions

We report here a comprehensive and exhaustive overview of the published literature on NLR as a predictive and/or prognostic factor in patients with BC. 

This analysis indicates that NLR is a reliable prognostic factor in localized BC treated with adjuvant chemotherapy. Indeed, in the 21/29 analyses (72.4%) where multivariate models were used, NLR was independently correlated with survival (DFS, BCSS and/or OS) [32,34,35,36,37,38,39,40,46,48,49,50,51,53]. 

Results are less clear-cut for patients with a localized disease receiving neo-adjuvant chemotherapy because only two of four studies (50%) found a correlation between NLR and DFS [25,30], and one of two studies (50%) found a correlation between NLR and OS [30]. The only study that analyzed NLR and BCSS found a significant correlation [25]. Conclusions on NLR and PCR in this population cannot be drawn because only one of two studies (50%) found a correlation between these factors [31]. 

Some heterogeneity in the NLR results in the neo-adjuvant chemotherapy setting could be linked to the different results observed in the different molecular subtypes (positive correlation only in patients with TNBC) and also to the small patient samples. Indeed, the total number of patients enrolled in the selected studies in this setting was 1558, compared with 18,153 patients in the adjuvant chemotherapy setting. 

Fewer data have been published on patients with metastatic BC. NLR appears to be a good prognostic factor for PFS and OS in this population, although studies are hard to compare due to the population heterogeneity [54,56,58,59,60,61] (i.e., BC molecular features, treatment type, number of previous chemotherapy courses). Therefore, no firm conclusion can be made on inflammatory blood markers as prognostic factors in patients with metastatic BC. 

Only very few studies focused on NLR or lymphopenia as predictive factors of chemotherapy-related toxicity [62,63,64,65]. Lymphopenia at day 1 of treatment was correlated with early death after chemotherapy [62]. However, no definitive conclusion could be made for patients with BC because the studied population included patients with various cancer types, and no specific subgroup analysis was performed in the BC subgroup. Lymphopenia at day 1 was inconsistently correlated with febrile neutropenia (two studies with opposite results [63,64]), and NLR was not correlated with peripheral neuropathy in patients receiving docetaxel [65]. Additional studies with larger populations should be performed to bring more data on NLR/lymphopenia as predictive factors of treatment toxicity. 

Due to its worse prognosis and higher chemosensitivity, TNBC was the most extensively evaluated molecular subgroup. Indeed, the results of the subgroup analyses on TNBC were reported in 10 studies [23,24,32,36,38,39,46,53,56]. Most of them (80%) found a correlation between NLR and survival in this subgroup [23,36,38,39,46,53,56], a much higher proportion of positive results than for the other BC subgroups, possibly related to TNBC-specific clinical history. 

Notably, most of the selected studies (33/41) evaluated Asian patients [23,25,28,30,31,32,35,36,38,39,42,43,44,46,47,48,49,51,53,54,55,56,58,59,61,64,65]. This demographic variable could be associated with specific molecular and pharmacological features that might lead to differences in the treatment toxicity and efficacy profiles. Another limitation concerns HER2-positive BC. Indeed, the use of anti-HER2 targeted therapy was inconsistently reported, leading to heterogeneity in survival predictions for this subgroup (not very clear). Considering these population-based sources of confusion, a large, multi-ethnic study should be carried out to evaluate, worldwide and for each world region, the prognostic and predictive value of NLR and other inflammatory blood markers in patients with BC.

Another limitation concerns the drugs used in the neo-adjuvant and adjuvant settings. Although most studies reported the use of anthracyclines and/or taxanes, only 47% of them (16/34) described the sequential or concomitant use of anthracycline- and taxane-based regimens, as recommended by the international learned societies. Data on chemotherapy regimens were missing in 53% of the studies (18/34).

Finally, all selected studies had a retrospective design. Although in most of them (>75%) at least one multivariate analysis was carried out, these results need to be validated in prospective studies or by retrospective evaluation in a prospective clinical trial. 

In conclusion, NLR appears to be a prognostic factor for DFS and OS in patients with early BC receiving adjuvant chemotherapy. Additional investigations in prospective studies would strengthen these results. On the other hand, the correlation between NLR and survival in neo-adjuvant chemotherapy settings remains unclear because most studies failed to show evidence of an independent correlation. Finally, most of the available data show that NLR is not a predictive factor for PCR in patients treated with neo-adjuvant chemotherapy; however, due to the population heterogeneity and/or small sample size of the published studies on this question, dedicated clinical trials are needed.

## Author Contributions

Conception and design: I.C., S.G.; Data analyses and interpretation: I.C., S.G. and W.J.; Drafting of the manuscript: I.C., S.G.; Revising critically the manuscript: I.C., S.G. and W.J.; Approval of the final manuscript: I.C., S.G. and W.J. All authors have read and agreed to the published version of the manuscript.

## Figures and Tables

**Figure 1 cancers-12-00958-f001:**
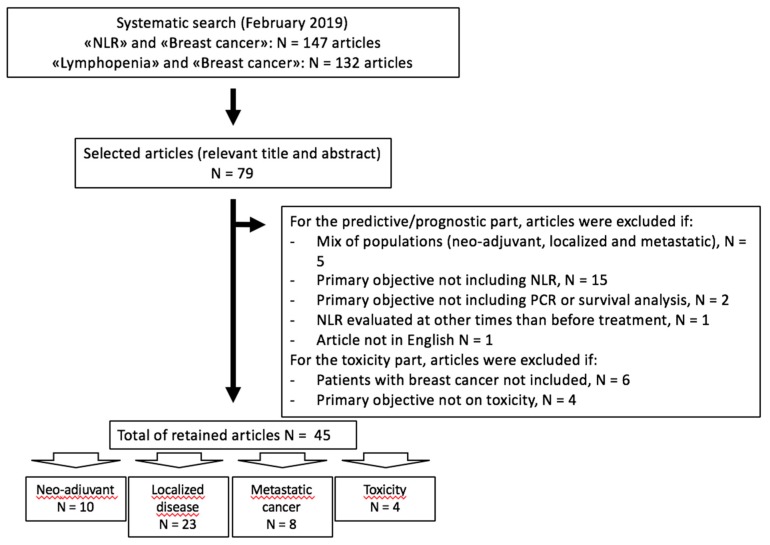
Flow chart of the study selection.

**Table 1 cancers-12-00958-t001:** Articles including data on neutrophil to lymphocyte ratio (NLR) as predictive and prognostic factor in patients with early breast cancer (BC) receiving neo-adjuvant chemotherapy.

First Author	Number of Patients	Treatment	Primary Objective	Cut-Off	Primary Objective Results (Univariate Analysis)	Results of Multivariate Models
Eryilmaz 2014 [22]	78 patients: all BC molecular subtypes	NS	NLR as predictive factor for PCR	2.33	−NLR and PCR	
Asano 2016 [23]	177 patients: 116 non TNBC (65.5%) 61 TNBC (34.5%)	Anthracyclines + taxanes	NLR as predictive and prognostic factor	3 (chosen before the statistical analysis)	−NLR and DFS (*p* = 0.849) −NLR and OS (*p* = 0.965) PCR was achieved in 28.6% of patients with high NLR vs 56.9% of patients with low NLR (*p* < 0.001)	
Suppan 2015 [24]	247 patients: 60.7% ER+ BC 54.3% PR+ BC 19.8% HER2+ BC	Anthracyclines + taxanes (58.3%); anthracyclines (38.2%); taxanes (2.8%); other (6.1%)	NLR as predictive and prognostic factor	Comparison of median NLR	−NLR and DFS (*p* = 0.363) −NLR and PCR (OR = 1.081; *p* = 0.053)	−NLR and DFS (HR = 1.01; *p* = 0.738)
Chen 2016 [25]	215 patients: 120 luminal A (55.8%) 52 luminal B (24.2%) 25 HER2+ (11.6%) 18 TNBC (8.4%)	Anthracyclines + taxanes (74.9%); anthracyclines (19.1%); taxanes (6%)	NLR as predictive and prognostic factor	2.1	NLR^low^ group showed higher PCR rate than NLR^high^ group (24.5% vs 14.3%; *p* < 0.05) +NLR and DFS (H = 2.11; *p* < 0.05) +NLR and BCSS (HR = 2.45; *p* < 0.05)	+NLR and DFS (HR = 1.57; *p* < 0.05) +NLR and BCSS (HR = 2.21; *p* < 0.05)
Marin-Hernandez 2017 [26]	150 patients: 32 luminal A (21.3%) 44 luminal B (29.4%) 35 HER2+ (23.3%) 39 TNBC (26%)	Anthracyclines + taxanes for all patients (except for 3 that received everolimus in the framework of a clinical trial)	Blood parameters as prognostic factors	3.33	+NLR and DFS (OR = 0.39; *p* = 0.019) +NLR and OS (OR = 0.38; *p* = 0.030)	−NLR and DFS (*p* = 0.154) −NLR and OS (*p* = 0.543)
Graziano 2019 [27]	373 patients 132 luminal A (35.4%) 44 luminal B/HER2− (11.8%) 69 luminal B/HER2+ (18.5%) 62 TNBC (16.6%) 66 HER2+ (17.7%)	Anthracyclines + taxanes (56.8%); anthracyclines or taxanes as single agents or in combination	NLR as predictive factor of PCR	2.42	−NLR and PCR (OR = 1.53; *p* = 0.125) −PLR and PCR (OR = 1.59; *p* = 0.084)	
Qian 2018 [28]	180 patients for PCR: 24 luminal A (13.3%) 60 luminal B (33.3%) 18 HER2+ positive (10%) 40 TNBC (22.2%) 38 not available (21.2%) 131 patients for survival	Taxane and/or anthracycline-based chemotherapy Only 40% of patients with HER2+ BC received trastuzumab	NLR/PLR as predictive and prognostic factors	2.44	+NLR and PCR (20% vs 7.8%; *p* = 0.030) survival analysis on 131 patients: −NLR and DFS (*p* = 0.535) or OS (data not available)	−NLR and PCR (*p* = 0.254)
Losada 2018 [29]	113 >65-year-old patients: 23 luminal A (20.4%) 57 luminal B (50.4%) 8 HER2+ (7.1%) 25 TNBC (22.1%)	Anthracycline, taxanes, or both (no specific data)	NLR and survival and PCR	3.33	−NLR and DFS (*p* = 0.42) or OS (*p* = 0.38) −NLR and PCR (*p* = 0.43)	
Koh 2014 [30]	157 patients with ER/PR+ and HER2− BC	Anthracyclines + taxanes (75.2%); anthracyclines (24.8%)	NLR as prognostic factor	2.25	+NLR and DFS (HR = 4.01; *p* = 0.001) +NLR and OS (HR = 24.64; *p* = 0.003)	+NLR and DFS (HR = 3.87; *p* = 0.002) +NLR and OS (HR = 24.87; *p* = 0.003)
Chae 2018 [31]	87 patients with TNBC	Anthracyclines + taxanes (71.3%); anthracyclines (28.7%)	NLR as predictive factor of PCR	1.7	Patients with low NLR had higher PCR rate (42.1% vs 18.4%; *p* = 0.018)	+NLR and PCR (OR = 4.27; *p* = 0.008)

NLR: neutrophil to lymphocyte ratio, PLR: platelet to lymphocyte ratio, PCR: pathological complete response, DFS: disease-free survival, OS: overall survival, ER: estrogen receptor, PR: progesterone receptor, BCSS: breast cancer-specific survival, OR: odd ratio, HR: hazard ratio, BC: breast cancer.

**Table 2 cancers-12-00958-t002:** Multivariate models (results and adjustment factors) for patients treated with neo-adjuvant chemotherapy.

All BC Molecular Subtypes					
Variable	PCR	DFS	OS	BCSS	Total
Number of multivariate models	1	3	1	1	6
Number of unique patients	180	612	150	215	1157
NLR significantly associated with *n* (%)	0 (0%)	1 (33%)	0 (0%)	1 (100%)	2 (33%)
Adjustment factors (%)					
Hormone receptors	100	67	NI	100	
T	NI	100	100	100	
N	NI	67	NI	100	
Age	NI	33	100	NI	
Histological grade	NI	33	NI	100	
Molecular subtype	100	NI	NI	NI	
Ki67	100	NI	NI	NI	
CRP	NI	33	NI	100	
Surgery method	NI	33	NI	100	
Lymphocyte count	100	33	100	NI	
Monocyte count	NI	33	100	NI	
Neutrophil count	NI	33	100	NI	
LMR	NI	33	100	NI	
NMR	NI	33	100	NI	
TNBC					
	PCR				Total
Number of multivariate models	1				1
Number of unique patients	87				87
NLR significantly associated with *n* (%)	1 (100%)				1 (100%)
Adjustment factors (%)					
Histological subtype	100				
Histological grade	100				
Ki67	100				
ER+ HER2- BC					
		DFS	OS		Total
Number of multivariate models		1	1		2
Number of unique patients		157	157		157
NLR significantly associated with *n* (%)		1 (100%)	1 (100%)		2 (100%)
Adjustment factors (%)					
PCR		100	100		
All studies					
	PCR	DFS	OS	BCSS	Total
Number of multivariate models	2	4	2	1	9
Number of unique patients	267	769	307	215	1558
NLR significantly associated with *n* (%)	1 (50%)	2 (50%)	1 (50%)	1 (100%)	5 (55.5%)

PCR: pathological complete response, DFS: disease free survival, OS: overall survival, BCSS: breast cancer specific survival, NLR: neutrophil to lymphocyte ratio, LMR: lymphocyte to monocyte ratio, NMR: neutrophil to monocyte ratio, NI: not indicated, T: tumor size, N: node invasion, CRP: C reactive protein.

**Table 3 cancers-12-00958-t003:** Articles including data on NLR as prognosis factor in patients with localized BC receiving adjuvant chemotherapy.

Author	Number of Patients	Treatment	Primary Objective	Cut-Off	Results for the Primary Objective (Univariate Analysis)	Results of Multivariate Models
Noh 2013 [32]	442 patients: 177 luminal A (48.7%) 69 luminal B (19.0%) 36 HER2+ (10.0%) 81 TNBC (22.3%)	NS	NLR as prognostic factor for DSS	2.5	+NLR and DSS 5-year survival: 88.6% *vs* 96.4%; 10-year survival: 84.3% *vs* 92.2%; *p* = 0.009	+NLR and BCSS (HR = 4.08; *p* = 0.003)
Cihan 2014 [33]	350 patients: 194 ER+ (55.4%) 183 PR+ (52.3%) 110 HER2+ (31.4%)	CT (94.3%) (based on anthracyclines for 71.7%)	NLR as prognostic factor for DFS and OS	3	−NLR and DFS (0R = 0.8; *p* = 0.410) −NLR and OS (OR = 0.7; *p* = 0.432)	
Forget 2014 [34]	720 patients: 601 ER+ (83.5%) 573 PR+ (79.6%) 67 HER2+ (9.3%)	NS	NLR as prognostic factor for DFS and OS	3.3	+NLR and DFS (HR = 2.20; *p* = 0.004) +NLR and OS (HR = 2.70; *p* = 0.020)	+NLR and DFS (HR = 1.99; *p* = 0.010) +NLR and OS (HR = 2.35; *p* = 0.046)
Nakano 2014 [35]	167 patients: 130 ER+ (77.8%) 93 PR+ (55.7%) 24 HER2+ (14.4%)	NS	NLR as prognostic factor for DFS and DSS	2.5 (according to previous studies)	+NLR and DFS (HR = 2.5; *p* = 0.004) +NLR and BCSS (HR = 3.2; *p* = 0.007)	−NLR and DFS (HR = 2.0; *p* = 0.070) +NLR and BCSS (HR=2.7; *p* = 0.045)
Yao 2014 [36]	608 patients: 330 luminal A (57.9%) 59 luminal B (10.3%) 83 HER2+ (14.6%) 98 TNBC (17.2%)	NS	NLR as prognostic factor for OS	2.57	−NLR and DFS (*p* = 0.084) +NLR and OS (*p* < 0.001)	+NLR and OS (RR = 3.63; *p* = 0.002)
Dirican 2015 [37]	1527 patients: 1019 ER+ (66.4%) 994 PR+ (64.7%) 249 HER2+ (16.2%)	Adjuvant CT (83.3%), NACT (9.6%)	NLR as prognostic factor for DFS and OS	4	+NLR and DFS (HR = 2.18; *p* < 0.001) +NLR and OS (HR = 2.82; *p* < 0.001)	+NLR and DFS (HR = 1.46; *p* = 0.028) +NLR and OS (HR = 1.91; *p* = 0.001)
Hong 2016 [38]	487 patients: 62 luminal A (12.7%) 244 luminal B (50.1%) 59 HER2+ (12.1%) 94 TNBC (19.3%) 28 NA (5.7%)	Adjuvant CT for 73.5% (anthracyclines 30.7%; taxanes 15.6%; anthracyclines + taxanes 36%; others 17.7%)	NLR as prognostic factor for DFS	1.93	+NLR and DFS (HR = 2.20; *p* = 0.002) −NLR and 5-year OS (90.8% *vs* 91.7%; *p* = 0.707)	+NLR and DFS (HR = 1.87; *p* = 0.011)
Jia 2015 [39]	1570 patients: 1001 luminal (63.8%) 344 HER2+ (21.9%) 225 TNBC (14.3%)	Adjuvant CT (85.4%)	NLR as prognostic factor for DFS and OS	2.0	+NLR and DFS (HR = 1.44; *p* = 0.005) +NLR and OS (HR = 1.58; *p* = 0.020)	+NLR and DFS (HR = 1.50; *p* = 0.004) +NLR and OS (HR = 1.63; *p* = 0.022)
**Author**	**Number of Patients**	**Treatment**	**Primary Objective**	**Cut-Off**	**Results for the Primary Objective (Univariate Analysis)**	**Results of Multivariate Models**
Orditura 2016 [40]	300 patients: 77 luminal A (25.7%) 124 luminal B HER2- (41.3%) 51 luminal B HER2+ (17%) 21 HER2-enriched (7%) 27 basal like (9%)	NS	NLR as prognostic factor for DFS	1.97	+NLR and DFS (HR = 0.45; *p* = 0.034)	+NLR and DFS (HR = 2.64; *p* = 0.013)
Ramos-Esquivel 2017 [41]	172 patients: 104 ER+ or PR+ and HER2- (60.5%) 18 ER+ or PR+ and HER2+ (10.5%) 16 ER− and PR- and HER2+ (9.3%) 34 ER− and PR- and HER2− (19.6%)	Adjuvant CT (83.1%), NACT (22.1%)	NLR as prognostic factor for DFS and OS	3	+NLR and DFS (HR = 4.20; *p* < 0.001) +NLR and OS (HR = 4.20; *p* < 0.001)	−NLR and DFS (HR = 1.97; *p* = 0.146) −NLR and OS (HR = 1.81; *p* = 0.192)
Zhang 2016 [42]	162 patients: 87 ER+ (53.7%) 77 PR+ (47.6%) 37 HER2+ (22.8%)	NS	NLR as prognostic factor for DFS	1.81 (according to the median value)	+NLR and DFS (HR = 1.81; *p* = 0.042) −NLR and OS	−NLR and DFS (HR = 1.43; *p* = 0.223)
Takeuchi 2017 [43]	296 patients: 253 ER+ (85%) 222 PR+ (75%) 247 HER2+ (83%)	Adjuvant CT according to the St Gallen recommendations	NLR as prognostic factor for DFS	2.06	−NLR and DFS	
Cho 2018 [44]	661 patients: 448 luminal (67.8%) 96 HER2+ (14.5%) 117 TNBC (17.7%)	NS	NLR as prognostic for DFS and DSS	1.34	+NLR and DFS (RR = 1.18; *p* < 0.001) +NLR and DSS (RR = 1.27; *p* < 0.001)	−NLR and DFS (RR = 1.24; *p* = 0.613) −NLR and BCSS (RR = 1.24; *p* = 0.681)
Ferroni 2018 [45]	475 patients: 164 luminal A (35%) 239 luminal B (50%) 15 HER2+ (3%) 57 TNBC (12%)	NACT (14.1%) adjuvant CT (82.5%) with anthracyclines	NLR as prognostic factor for DFS and OS	2	+NLR and DFS (HR= 2.28; *p* = 0.001) +NLR and OS (HR = 3.39; *p* = 0.050)	
Geng 2018 [46]	1374 patients in the testing group: 1038 Hormone receptor+ (75.6%) 336 Hormone receptor− (24.4%) 128 HER2+ (9.3%) 1246 HER2− (90.7%) 208 TNBC (15.1%) 1166 No TNBC (84.9%)	96 patients in cohort 1 received NACT	NLR as prognostic factor for DFS	1.878 (in the testing group)	+NLR and DFS testing group (HR = 2.89; *p* < 0.001)	+NLR and DFS in the testing group (HR = 2.99; *p* < 0.001)
**Author**	**Number of Patients**	**Treatment**	**Primary Objective**	**Cut-Off**	**Results for the Primary Objective (Univariate Analysis)**	**Results of Multivariate Models**
Geng 2018 [46]	1084 patients in the validation group: 702 Hormone receptor+ (64.7%) 382 Hormone receptor− (35.3%) 170 HER2+ (15.7%) 914 HER2− (84.3%) 212 TNBC (19.6%) 872 No TNBC (80.4%)	NS	NLR as prognostic factor for DFS	1.878 (based on the testing group)	+NLR and DFS in the validation group (HR = 1.65; *p* = 0.017)	+NLR and DFS in the validation group (HR = 1.64; *p* = 0.023)
Fujimoto 2018 [47]	889 patients: 699 ER+ (78.6%) 152 HER2+ (17.1%)	Adjuvant CT (29.6%)	NLR as prognostic factor for DFS	2.72	+NLR and DFS (HR = 1.56; *p* = 0.047) −NLR and OS (*p* = 0.23)	−NLR and DFS (*p* = 0.14)
Kim 2016 [48]	220 patients with pN3 BC: 99 Hormone receptor+/HER2− (45%) 44 Hormone receptor+/HER2+ (20%) 48 Hormone receptor−/HER2+ (21.8%) 29 TNBC (13.2%)	Adjuvant CT (anthracyclines followed by taxanes) for all patients	NLR as prognostic factor for DFS	3 (from previous studies)	+NLR and 5-year DFS (*p* = 0.043)	+NLR and DFS (HR = 3.93; *p* = 0.020)
Qiu 2018 [49]	406 patients with TNBC	NACT (21.2%) Adjuvant CT (78.8%)	NLR as prognostic factor for DFS and OS	2.85	+NLR and DFS (HR = 2.63; *p* < 0.001) +NLR and OS (HR = 3.26; *p* < 0.001)	+NLR and DFS (HR = 2.13; *p* = 0.008) +NLR and OS (HR = 2.69; *p* = 0.001)
Pistelli 2015 [50]	90 patients with TNBC	NS	NLR as prognostic factor for DFS	3	+NLR and DFS (*p* = 0.002) +NLR and OS (*p* = 0.003)	+NLR and DFS (HR = 5.15; *p* = 0.03) +NLR and OS (HR = 6.16; *p* = 0.01)
Lee 2019 [51]	358 patients with TNBC	Adjuvant CT (86.6%): anthracyclines (50.9%), anthracyclines + taxanes (22.4%), others (26.7%). NACT (14%): anthracyclines + taxanes (64%), anthracyclines (36%)	NLR as prognostic factor for DFS and OS	3.16	+NLR and DFS (HR = 2.11; *p* = 0.036) +NLR and OS (HR = 2.97; *p* = 0.003)	−NLR and DFS (*p* = 0.14) +NLR and OS (HR = 3.15; *p* = 0.009)
Patel 2019 [52]	126 patients with TNBC	NACT (31.7%), adjuvant CT (52.4%), or both (4.8%)	NLR as prognostic factor for DFS and OS	NLR: 3 (based on previous studies)	−Baseline NLR and DFS (*p* = 0.77) −Baseline NLR and OS (*p* = 0.23)	
Liu 2016 [53]	318 patients with hormone receptor-negative BC: 157 HER2+ (49.4%) 161 HER2− (50.6%)	Adjuvant CT (81.5%), NACT (17.6%), none (0.9%)	NLR as prognostic factor for DFS and OS	3	+NLR and DFS (HR = 2.37; *p* < 0.001) +NLR and OS (HR = 3.09; *p* < 0.001)	+NLR and DFS (HR = 1.89; *p* < 0.001) +NLR and OS (HR = 3.09; *p* < 0.001)

NLR: neutrophil to lymphocyte ratio, PLR: platelet to lymphocyte ratio, PCR: pathological complete response, DFS: disease free survival, OS: overall survival, CT: chemotherapy, NACT: neoadjuvant chemotherapy, ER: estrogen receptor, PR: progesterone receptor, BCSS: breast cancer specific survival, OR: odd ratio, HR: hazard ratio, BC: breast cancer, TNBC: triple negative breast cancer.

**Table 4 cancers-12-00958-t004:** Multivariate models (results and adjustment factors) for patients with localized BC receiving adjuvant treatment.

All BC Molecular Subtypes				
Variable	DFS	OS	BCSS	Total
Number of multivariate models	13	5	3	21
Number of unique patients	9333	4597	1879	15809
NLR significantly associated with *n* (%)	8 (61.5%)	4 (80%)	2 (66%)	14 (66%)
Adjustment factor (%)				
T	77	80	33	
N	70	60	100	
AJCC stage	38	40	NI	
Age	31	20	67	
Menopausal status	23	NI	NI	
Hormone receptors	23	NI	67	
HER 2 status	8	NI	NI	
Molecular subtype	54	80	NI	
Histological grade	38	20	NI	
LVI	8	NI	33	
Perineural invasion	NI	NI	33	
Ki67	8	NI	NI	
Multiplicity	8	NI	NI	
Adjuvant chemotherapy	15	20	NI	
Endocrine therapy	8	NI	NI	
Use of NSAIDs	8	20	NI	
PLR	15	40	33	
LMR	15	20	33	
MCH	8	NI	NI	
RDW	NI	20	NI	
dNLR	8	NI	33	
TNBC				
	DFS	OS		Total
Number of multivariate models	3	3		6
Number of unique patients	854	854		1708
NLR significantly associated with *n* (%)	2 (66%)	3 (100%)		5 (83%)
Adjustment factor (%)				
T	67	67		
N	67	67		
AJCC stage	33	33		
Age	100	67		
Menopausal status	33	33		
Histological subtype	33	33		
Histological grade	67	67		
Ki67	33	67		
Necrosis	33	33		
LVI	67	67		
Type of surgery	33	33		
Adjuvant chemotherapy (vs NACT)	33	33		
Adjuvant radiotherapy	33	33		
Cancer recurrence	NI	33		
Hormone receptor-negative BC				
	DFS	OS		Total
Number of multivariate models	1	1		2
Number of unique patients	318	318		636
NLR significantly associated with *n* (%)	1 (100%)	1 (100%)		2 (100%)
Adjustment factor (%)				
T	100	100		
N	100	100		
Age	100	100		
Histological grade	100	100		
HER 2 status	100	100		
PLR	100	100		
All studies				
	DFS	OS	BCSS	Total
Number of multivariate models	17	9	3	29
Number of unique patients	10505	5769	1879	18153
NLR significantly associated with *n* (%)	11 (65%)	8 (89%)	2 (66%)	21 (72.4%)

CT: chemotherapy, DFS: disease free survival, OS: overall survival, BCSS: breast cancer specific survival, NLR: neutrophil to lymphocyte ratio, NSAIDs: non-steroidal anti-inflammatory drugs, LMR: lymphocyte to monocyte ratio, MCH: mean corpuscular hemoglobin, NMR: neutrophil to monocyte ratio, RDW: red cell distribution width, dNLR: derived NLR, PLR: platelet to lymphocyte ratio, LVI: lympho-vascular invasion, NACT: neoadjuvant chemotherapy, NI: not indicated.

**Table 5 cancers-12-00958-t005:** Articles with data on NLR as prognostic factor in patients with metastatic breast cancer.

Author	Number of Patients	Treatment	Primary Objective	Cut-Off	Results for the Primary Objectives (Univariate Analysis)	Results of Multivariate Models
Iwase 2017 [54]	89 patients with recurrent BC after surgery: 31 ER+ HER2− (35%) 20 ER+ HER2+ (22%) 14 HER2 type (16%) 24 TNBC (27%)	NS	NLR and prognosis	3 (based on previous studies)	+NLR and OS (HR = 2.68; *p* < 0.05)	+NLR and OS (HR = 2.93; *p* > 0.05)
Araki 2018 [55]	51 patients with HER2+ BC: 14 ER+ (47%) 5 PR+ (17%)	Pertuzumab and trastuzumab combined with eribulin (ERI) (*n* = 30) or nab-paclitaxel (*n* = 21)	Blood-based prognostic parameters	2 (median value)	−NLR and PFS	
Miyagawa 2018 [56]	85 patients: 62 ER+ (73%) 39 PR+ (46%) 4 HER2+ (5%)	Eribulin (*n* = 59) or nab-paclitaxel (*n* = 26)	NLR and prognosis according to the treatment	3 (based on previous studies)	+NLR and PFS in the eribulin group (HR = 0.37; *p* = 0.003) −NLR and PFS in the nab-paclitaxel group (*p* = 0.84) −NLR and OS in the eribulin group (*p* = 0.058) −NLR and OS in the nab-paclitaxel group (*p* = 0.15)	+NLR and PFS in the eribulin group (HR = 0.39; *p* = 0.007)
De Sanctis 2018 [57]	71 patients: 53 hormone receptor+ (75%) 8 HER2+ (11%) 11 TNBC (15%)	Eribulin (after 2 to 5 previous lines of chemotherapy)	NLR as prognostic factor	2.5–4–5.5	−NLR and PFS (*p =* 0.5) for any cut-off value	
Takuwa 2018 [58]	171 patients: 93 ER+ HER2− (54.4%) 23 ER+ HER2+ (13.5%) 20 ER− HER2+ (11.7%) 28 ER− HER2− (16.4%) 7 unknown (4.0%)	NS	NLR as prognostic factor	1.9	+NLR and OS (33 vs 79 months, *p* = 0.004)	+NLR and OS (HR = 1.75; *p* = 0.022)
Author	Number of patients	Treatment	Primary objective	Cut-off	Results for the primary objectives (univariate analysis)	Results of multivariate models
Iimori 2018 [59]	34 patients receiving ET as initial drug therapy: 4 HER2+ (12%)	Endocrine therapy: letrozole (58.8%); anastrozole (20.6%); tamoxifen (with/without LHRH) (17.7%); exemestane (2.9%)	NLR as predictive factor of the response to endocrine therapy and prognosis	3 (based on previous studies)	+NLR and PFS (HR = 3.94; *p* = 0.016) +NLR and OS (*p* = 0.013) +NLR and time to treatment failure (*p* = 0.031)	+NLR and PFS (HR = 3.93; *p* = 0.008)
Vernieri 2018 [60]	57 patients with TNBC	Platinum-based chemotherapy: carboplatin-paclitaxel (84%) or carboplatin-gemcitabine (16%); first line (88%) or second line (12%)	NLR as prognostic factor	2.5 (based on previous studies)	+NLR and PFS (HR = 3.25; *p* < 0.001)	+NLR and PFS (HR = 2.65; *p* = 0.004)
Imamura 2019 [61]	53 patients with HER2+ BC	TDM-1	NLR as prognostic factor	2.56	+NLR and PFS (HR = 0.23; *p* < 0.001) +NLR and OS (HR = 0.38; *p* = 0.0296)	+NLR and PFS (HR = 0.27; *p* = 0.0019) +NLR and OS (HR = 0.35; *p* = 0.018)

NLR: neutrophil to lymphocyte ratio, PLR: platelet to lymphocyte ratio, PCR: pathological complete response, DFS: disease-free survival, OS: overall survival, ER: estrogen receptor, PR: progesterone receptor, BCSS: breast cancer-specific survival, OR: odd ratio, HR: hazard ratio, BC: breast cancer, ET: endocrine therapy, TNBC: triple negative breast cancer.

**Table 6 cancers-12-00958-t006:** Multivariate models (results and adjustment factors) for patients with metastatic breast cancer.

All BC Molecular Subtypes			
Variable	PFS	OS	Total
Number of multivariate models	1	2	3
Number of unique patients	85	260	345
NLR significantly associated with n (%)	1 (100%)	2 (100%)	3 (100%)
Adjustment factor (%)			
Menopausal status	NI	50	
BMI	NI	50	
Molecular subtype	NI	50	
LDH	NI	50	
Complete response	NI	50	
Primary tumor stage IV	NI	50	
Number of metastatic sites	NI	50	
Visceral metastasis sites (≥2 vs <2)	NI	50	
Previous chemotherapy	100	NI	
Hormone receptor-positive BC			
	PFS		Total
Number of multivariate models	1		1
Number of unique patients	34		34
NLR significantly associated with *n* (%)	1 (100%)		1 (100%)
Adjustment factor (%)			
Objective response to endocrine therapy	100	NI	
TNBC			
	PFS		Total
Number of multivariate models	1		1
Number of unique patients	57		57
NLR significantly associated with n (%)	1 (100%)		1 (100%)
Adjustment factor (%)			
Visceral metastases	100	NI	
Maintenance chemotherapy	100	NI	
Previous exposure to taxanes	100	NI	
PLR	100	NI	
HER 2+ BC			
	PFS	OS	Total
Number of multivariate models	1	1	2
Number of unique patients	53	53	106
NLR significantly associated with *n* (%)	1 (100%)	1 (100%)	2 (100%)
Adjustment factor (%)			
Disease control during 1^st^ line therapy	100	100	
Number of metastatic sites	100	100	
All patients			
	PFS	OS	Total
Number of multivariate models	4	3	7
Number of unique patients	229	313	542
NLR significantly associated with *n* (%)	4 (100%)	3 (100%)	7 (100%)

PFS: progression-free survival, OS: overall survival, NLR: neutrophil to lymphocyte ratio, PLR: platelet to lymphocyte ratio, BMI: body mass index, NI: not indicated, LDH: dehydrogenase lactate.

**Table 7 cancers-12-00958-t007:** Articles with data on inflammatory blood markers as predictive factor of toxicity in patients with breast cancer.

Author	Number of Patients	Population of Interest	Treatment	Primary Objective	Cut-Off	Results for the Primary Objective (Univariate Results)	Results of Multivariate Models
Ray Coquard 2001 [62]	1051	First-line chemotherapy (BC, colon, ovary, head and neck, lung, other cancer type)	NS	To establish a risk model for early death after chemotherapy (defined as death within 1 month after treatment administration)	Lymphocytes <0.7 G/L	Predictive of early death: day 1 lymphocyte count <0.7 G/L (*p* < 0.01) day 1 platelet count <150 G/L (*p* = 0.01)	Predictive of early death: day 1 lymphocytes < 0.7 G/L (OR = 3.1)
Ray Coquard 2003 [63]	3 groups: −950 (CLM-1996 cohort) −321 (Elypse 1 cohort) −329 (Elypse 0 cohort)	All cancers (BC, colon-rectum, ovary, head and neck, lung, lymphoma, myeloma, sarcoma, germ cell tumors, other) treated by chemotherapy (regardless of previous treatments)	NS	To evaluate a risk model for FN using only day 1 blood cell count, and to compare the day 1 and day 5 risk models	Lymphocytes <0.7 G/L	In the CLB-1996 cohort: +lymphocytes at day 1 and FN (*p* = 0.05) Lymphocytes at day 5 and FN data not available in the Elypse 1 cohort: −lymphocytes at day 1 and FN (*p* = 0.18) +lymphocytes at day 5 and FN (*p* < 0.01) In the Elypse 0 cohort: −lymphocytes at day 1 and FN (*p* = 0.08) +lymphocytes at day 5 and FN (*p* < 0.01)	Lymphocytes at day 1 and FN (OR = 1.75; *p* = 0.02)
Choi 2003 [64]	82	All cancers (non-Hodgkin lymphoma, stomach, BC, NSCLC, hepatobiliary, sarcoma, colorectal cancer and others) receiving first course of chemotherapy	NS	To evaluate lymphocyte count at day 1, day 3 and day 5 as a way to identify patients at risk of FN	Lymphocytes <0.7 G/L and lymphocytes <0.5 G/L	For lymphocytes ≤0.5 G/L: −day 1 and FN (*p* = 0.33) +day 3 and FN (*p* < 0.01) +day 5 and FN (*p* = 0.023) For lymphocytes ≤ 0.7 G/L: −day 1 and FN (*p* = 0.05) +day 3 and FN (*p* = 0.01) +day 5 and FN (*p* < 0.01)	Day 5 lymphocytes ≤ 0.7 G/L and NF (OR = 19.0 *p* = 0.01)
Yamanouchi 2017 [65]	67	BC, all stages (only 6% stage IV)	Docetaxel 75 mg/m^2^ at least 4 cycles	To elucidate the relationship between PN and NLR, PLR and MLR	Median NLR in patients with or without toxicity	No correlation between NLR, PLR, or MLR before or at the first or third cycle and PN occurrence	

NLR: neutrophil to lymphocyte ratio, PLR: platelet to lymphocyte ratio, MLR: monocyte to lymphocyte ratio, OR: odd ratio, BC: breast cancer, FN: febrile neutropenia, PN: peripheral neuropathy.

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
