# Peer review of "Neutrophil to Lymphocyte Ratio as Prognostic and Predictive Factor in Breast Cancer Patients: A Systematic Review"

_cancers, 2020, doi:10.3390/cancers12040958_

Round 1

Reviewer 1 Report

The paper of Corbeau et al, is a systematic review reporting an overview of published works evaluating neutrophil to lymphocyte ratio as prognostic or predictive factor of pathological complete response and toxicity in early and advanced breast cancer.

The paper attempt to identify articles published according to selection criteria related to the search of specific terms on Pubmed database. The search is updated on February 2019.

This review provides a list of articles grouped per results (on PCR, on DFS, on OS and BCSS) and including studies for all molecular subtypes or specific molecular subtypes. Seven informative tables are provided. The limit of this work is that does not include additional advancements, i.e. try to make statistical analyses of data of papers with similar results/information.

Major revision are below provided:

INTRODUCTION

The first sentence “Brest cancer (BC) is …. subtype” needs revision. Other sentences and English should be revised.

The sentence “limphocytes belonging to the innate arm of the immune system” is too generic, please improve.

RESULTS

Pag 5 not “2.1.1.1.2.3.” but “3.1.1.1.2.3.” should be revised.

Pag7.TABLE1:3rd box , the sum of patients “247” is not correct please revise.

Pag8 TABLE1: LOSADA 2018, the sum of patients “104” is not correct please revise.

TABLE 3: format should be revised.

Author Response

*Review 1

The paper of Corbeau et al, is a systematic review reporting an overview of published works evaluating neutrophil to lymphocyte ratio as prognostic or predictive factor of pathological complete response and toxicity in early and advanced breast cancer.

The paper attempt to identify articles published according to selection criteria related to the search of specific terms on Pubmed database. The search is updated on February 2019.

This review provides a list of articles grouped per results (on PCR, on DFS, on OS and BCSS) and including studies for all molecular subtypes or specific molecular subtypes. Seven informative tables are provided. The limit of this work is that does not include additional advancements, i.e. try to make statistical analyses of data of papers with similar results/information.

 Major revision are below provided:

INTRODUCTION

The first sentence “Brest cancer (BC) is …. subtype” needs revision. Other sentences and English should be revised.

Response: The first sentence has been changed:“Breast cancer (BC) prognosis depends not only on the tumor stage (localized versusmetastatic disease), but also on the molecular subtype (luminal, HER2+, or triple-negative BC).”

 English language has been revised.

The sentence “lymphocytes belonging to the innate arm of the immune system” is too generic, please improve.

Response:The text has been improved. “In recent years, the role of tumor infiltrating lymphocytes (TILs), especially in BC, also has been studied [12]. TILsare a selected population of T cells that show high specific immunological reactivity against tumor cells.These lymphocytes, which are part of the innate immune system, can detect cancer cells and alert the immune system that will destroy them.”

RESULTS

Pag 5 not “2.1.1.1.2.3.” but “3.1.1.1.2.3.” should be revised.

Response: The correction was made.

Pag7.TABLE1:3rd box , the sum of patients “247” is not correct please revise.

Response: The sum is correct, this study included 247 patients; 60.7% with ER+,  54.3% with PR+, and 19.8% with HER2+ breast cancer. The sentence has been changed for more clarity.

Pag8 TABLE1: LOSADA 2018, the sum of patients “104” is not correct please revise

Response: the correction was made: N = 113 patients.

TABLE 3: format should be revised.

Response: The Table 3 has been revised.

Reviewer 2 Report

The review entitled "Neutrophil to lymphocyte ratio as prognostic and
predictive factor in breast cancer patients: a systematic review" is a well written and extensive overview of the published literature that examines NLR as a prognostic factor in BC patients receiving adjuvant treatment, predictive and prognostic factor in patients receiving NACT and as prognosis factor in metastatic breast cancer patients.

This review is timely as the ability of NLR to predict clinical response to a range of treatments is currently being extensively investigated across tumour types.

A minor amendment noted:

In section 2.3 "data extraction" the authors should clarify whether two or more reviewers independently extracted the useful data from the eligible studies.

Refs 42 & 59 change text authors to lower case

Author Response

*Review 2

The review entitled "Neutrophil to lymphocyte ratio as prognostic and
predictive factor in breast cancer patients: a systematic review" is a well written and extensive overview of the published literature that examines NLR as a prognostic factor in BC patients receiving adjuvant treatment, predictive and prognostic factor in patients receiving NACT and as prognosis factor in metastatic breast cancer patients.

This review is timely as the ability of NLR to predict clinical response to a range of treatments is currently being extensively investigated across tumour types.

A minor amendment noted:

In section 2.3 "data extraction" the authors should clarify whether two or more reviewers independently extracted the useful data from the eligible studies.

Response: A sentence has been added to clarify this point: “Two reviewers independently extracted the following data from the selected studies...”

Refs 42 & 59 change text authors to lower case

Response: References have been changed.

Reviewer 3 Report

This systematic review provides an efficient, comprehensive and important summary of the use of inflammatory biomarkers in breast cancer prognosis.

There are a few modifications that are required or will improve the manuscript.

For the tables, particularly tables 3, 5 and 7 it would benefit the reader to have the references in their numeric form included (in column 1) in addition to the authors reference.

Within the methods this same search term appears twice ‘lymphopaenia’ AND ‘breast cancer’ – if this was specifically for the ‘toxicity’ assessment in the second instance then that is not clear as written.

There are many points where the English language is not as clear as it could be and an extensive revision of this (particularly in the results and conclusion) should be conducted. The following 2 sentences must be addressed as it is very unclear what they are saying

Last sentence pg 15 They found a correlation between NLR and OS as well in univariate than in multivariate analyses (HR = 3.09; 95%CI 2.35-4.06; p < 0.001).’

‘In 2001, Ray Coquard et al [62] studied predictive factors for early death after CT (defined as death within one month after the administration of cancer CT) in a prospective study and found, in a group of 1051 patients with different types of cancer (including 756 [33%] BC patients), that lymphocyte count at day 1 < 0.7 G/L was predictive for early death cancer as well in univariate than in multivariate models (OR = 3.1 ; 95%CI 1.8-5.8; p < 0.001)’ - This sentence is both very long and does not make sense as written.

Author Response

*Review 3

This systematic review provides an efficient, comprehensive and important summary of the use of inflammatory biomarkers in breast cancer prognosis.

There are a few modifications that are required or will improve the manuscript.

For the tables, particularly tables 3, 5 and 7 it would benefit the reader to have the references in their numeric form included (in column 1) in addition to the authors reference.

Response: The references in their numeric form have been included in column 1 in Tables 3, 5 and 7.

 Within the methods this same search term appears twice ‘lymphopaenia’ AND ‘breast cancer’ – if this was specifically for the ‘toxicity’ assessment in the second instance then that is not clear as written.

 Response: These sentences have been changed for more clarity: “We performed a systematic search of the PubMed database using the following search terms: “neutrophil to lymphocytes ratio” or “lymphopenia” AND “breast cancer”. We also looked for articles using the search terms “toxicity” AND “neutrophil to lymphocyte ratio” AND “breast cancer” and “toxicity” AND “lymphopenia” AND “breast cancer”.”

There are many points where the English language is not as clear as it could be and an extensive revision of this (particularly in the results and conclusion) should be conducted.

Response: English language has been revised.

The following 2 sentences must be addressed as it is very unclear what they are saying

Last sentence pg 15 ‘They found a correlation between NLR and OS as well in univariate than in multivariate analyses (HR = 3.09; 95%CI 2.35-4.06; < 0.001).’

Response: The sentence has been revised: “Liu et al[53]found that in patients with hormone receptor-negative BC, high NLR was associated with poor OS in univariate (HR = 3.09; 95%CI 2.35-4.06; p <0.001) and multivariate analyses (HR = 3.09; 95%CI 2.35-4.06; p<0.001).”

‘In 2001, Ray Coquard et al [62] studied predictive factors for early death after CT (defined as death within one month after the administration of cancer CT) in a prospective study and found, in a group of 1051 patients with different types of cancer (including 756 [33%] BC patients), that lymphocyte count at day 1 < 0.7 G/L was predictive for early death cancer as well in univariate than in multivariate models (OR = 3.1 ; 95%CI 1.8-5.8; < 0.001)’ - This sentence is both very long and does not make sense as written.

Response: This paragraph has been changed for more clarity: “Ray Coquard et al[62]evaluated predictive factors for early death after chemotherapy (defined as death within one month after the administration of cancer treatment) in a prospective study. They included 1051 patients among whom 756 (33%) had BC. They found that lymphopenia was a predictive factor of worse survival, in univariate and multivariate models (OR = 3.1; 95%CI 1.8-5.8;p<0.001).”

Reviewer 4 Report

It will be useful to have a tabulated list of abbreviations for references

These are some minor corrections in sentence formation and grammar

Author Response

*Review 4

Comments and Suggestions for Authors

It will be useful to have a tabulated list of abbreviations for references

Response: A list of abbreviations has been added.

These are some minor corrections in sentence formation and grammar

Response: English language has been revised.

Round 2

Reviewer 1 Report

The authors addressed the requirements.